# Modifications to Infant Formula Instructions Improve the Accuracy of Formula Dispensing

**DOI:** 10.3390/nu12041150

**Published:** 2020-04-20

**Authors:** Linda A. Gilmore, Abby D. Altazan, Emily W. Flanagan, Alexandra G. Beyer, Kelsey N. Olson, Alexis A. O’Connell, Timothy H. Nguyen, Robbie A. Beyl, Leanne M. Redman

**Affiliations:** Pennington Biomedical Research Center, Baton Rouge, LA 70808, USA; anne.gilmore@pbrc.edu (L.A.G.); abby.altazan@pbrc.edu (A.D.A.); emily.flanagan@pbrc.edu (E.W.F.); agbeyer1@comcast.net (A.G.B.); Kelsey.Olson@pbrc.edu (K.N.O.); oconnella@acom.edu (A.A.O.); timothynguyen@outlook.com (T.H.N.); Robbie.Beyl@pbrc.edu (R.A.B.)

**Keywords:** feeding, infant formula, infant growth

## Abstract

Readability of infant formula preparation instructions is universally poor, which may result in inaccurate infant feeding. Given that inaccurate formula dispensing can lead to altered infant growth and increased adiposity, there is an increased need for easy to follow instructions for formula preparation. We hypothesize that altering infant formula instruction labels using feedback from iterative focus groups will improve the preparation accuracy of powdered infant formula in a randomized controlled trial. Participants were recruited from the community, 18 years of age or older, willing to disclose demographic information for focus group matching, and willing to participate freely in the first (*n* = 21) or second (*n* = 150) phase of the study. In the second phase, participants were randomized to use the standard manufacturer instructions or to use the modified instructions created in the first phase. Accuracy was defined as the percent error between manufacturer-intended powder formula quantity and the amount dispensed by the participant. Participants who were assigned to the modified instructions were able to dispense the powdered formula more accurately than participants who used the standard manufacturer instructions (−0.67 ± 0.76 vs. −4.66 ± 0.74% error; *p* < 0.0001). Accuracy in powdered formula dispensing was influenced by bottle size (*p* = 0.02) but not by body mass index (*p* = 0.17), education level (*p* = 0.75), income (*p* = 0.7), age (*p* = 0.89) or caregiver status (*p* = 0.18). Percent error of water measurement was not different between the groups (standard: −1.4 ± 0.6 vs. modified: 0.7 ± 0.6%; *p* = 0.38). Thus, caloric density was more accurate in the modified instructions group compared to the standard manufacturer instructions group (−0.3 ± 0.6 vs.−2.9 ± 0.9%; *p* = 0.03). Infant formula label modifications using focus group feedback increased infant formula preparation accuracy.

## 1. Introduction

The World Health Organization and the American Academy of Pediatrics recommend exclusive breastfeeding until six months of age; however, over 80% of infants in the United States receive infant formula prior to their six month birthday [1,2]. Despite the high prevalence of formula feeding and the critical nature of nutrition during this developmental period, most caregivers receive little instruction on proper infant formula preparation and infant feeding [3,4], which may lead to inaccurate nutrition provision [5,6,7,8] and altered infant growth [8,9,10,11,12]. While efforts are taken by formula manufacturers to mimic the nutritional profile of breast milk, formula-fed infants have higher rates of growth over their first year [13]. We have previously proposed that inaccurate formula dispensing may contribute to the rapid weight gain and increased adiposity in formula-fed infants [8].

To prepare a bottle of infant formula, caregivers are required to correctly interpret the package label to accurately estimate the serving size of both powder and water. Difficulty in understanding preparation instructions together with an inaccurate measurement of formula can lead to incorrect preparation of an infant formula bottle. Indeed, infant formula preparation instructions have been criticized for having poor readability and an average reading difficulty at the college level [3]. Infant formula preparation instructions may therefore benefit from being re-designed to explain formula dispensing more clearly. Participatory research may help to improve label comprehension and competency in bottle preparation. Participatory research with caregivers provides feedback based on personal experiences while research with non-caregivers allows non-biased, raw feedback to increase preparation accuracy to ensure proper feeding of infants.

Recommendations have been made to improve the accuracy of medication dispensing through label and instrument modifications and advanced medication counseling [14,15,16]. However, despite the need to improve infant formula instructions to improve the accuracy of infant feeding and thereby optimize healthy infant growth [3], to our knowledge, no studies have tested and modified instructions. Using a participatory research model followed by a randomized controlled trial, the objective of this work was to improve the accuracy by which individuals measure powdered infant formula by providing greater clarity of the preparation instructions.

## 2. Materials and Methods

### 2.1. Participants and Methods

The study was approved by the Institutional Review Board and was registered as a clinical trial named Unintentional Overfeeding of Formula-Fed Infants (Whoa Baby, NCT02701868). Individuals from the participatory research and the randomized controlled trial were recruited through standard recruitment methods of Pennington Biomedical Research Center, including descriptions placed on the research center webpage, database emails sent to individuals registered to be notified about upcoming research studies, social media advertisements directed to residents of the Greater Baton Rouge area, and printed flyers placed around town. Recruitment materials advertised a research study testing how modified powdered infant formula instructions would improve formula preparation. Informed consent was obtained from each participant individually before the initiation of any study procedures, and compensation was provided for completion of study procedures.

### 2.2. First Phase—Participatory Research

#### 2.2.1. Focus Group Participants

Eligibility criteria were greater than or equal to 18 years of age, willing to disclose demographic information for focus group matching, willing to participate freely in the focus group, and be available for the designated meeting time, since focus groups were scheduled based off staff and space availability. The focus groups were completed between August and October 2016. Within each focus group, participants were grouped in order to achieve similar demographic representation (level of education, household income, and relation to infants) and caregiver status to encourage equal participation of attendees. Caregiver status was defined as: primary caregiver—routine care for an infant in the last year for more than 12 h per day; secondary caregiver—routine care for an infant in the last year for less than 12 h per day; non-caregiver—had not provided routine care for an infant in the last year. Five focus group sessions were conducted with 21 participants (Figure 1).

#### 2.2.2. Design

Iterative focus groups were used to evaluate and provide recommendations to improve the reading and comprehension of powdered infant formula package preparation instructions. In focus group discussions, individuals from different backgrounds and experiences with infant formula provided input to barriers and facilitators of bottle preparation and offered recommendations to increase comprehension of the instructions.

#### 2.2.3. Measures

Non-fasting body weight was measured in light clothing on a calibrated scale (GSE, Livonia, MI, USA), height was measured using a wall mounted stadiometer (Holtain Limited, Crymych, UK) without shoes in a private clinic room and body mass index (BMI) was calculated. Demographic information and experience with infant formula preparation was collected in a structured interview. To familiarize participants with the bottle-making process, participants independently prepared two, 2 fl oz bottles of infant formula using standard manufacturer instructions provided in a research demonstration kitchen at Pennington Biomedical Research Center. The first bottle was prepared without assistance or interruption and the second bottle preparation was interrupted at each step to allow for the researcher to weigh the dry powder and the added water. These data were collected to inform the participant during discussions, but not included in the second-phase data analysis. Focus group discussions were initiated by a moderator who adhered to a script of group discussion probes. Participants were asked to comment on discussion questions (Figure 2) and to provide recommendations for improving formula preparation instructions.

Prior to the first focus group, the standard manufacturer instructions were modified by the study team with the goal to increase label comprehension. These modified instructions were presented to the first focus group for discussion. Instructions were then modified in a stepwise fashion via iterative focus groups such that the modified instructions resulting from the previous focus group were shown and discussed with the following focus group. The final focus group (Group 5) was asked the same questions as groups 1 through 4 and, with no further recommendations, approved the final modified instructions. Focus group discussions were recorded via audio tape and transcribed for analysis by two independent coders using NVivo 11 (QSR International, Melbourne, Australia) according to a coding scheme developed by the study team prior to analysis. Coding analysis focused on the label-change recommendations.

### 2.3. Second Phase—Double-Blind Randomized, Controlled Trial

#### 2.3.1. Design

The second phase was a double-blind, randomized controlled trial to test the accuracy of infant formula preparation with the instructions modified in the first phase compared to the original manufacturer instructions. Individuals were randomized equally to one of two groups: modified instructions or standard manufacturer instructions. To enable blinding, both sets of preparation instructions were scaled and fixed to a commercially available formula container that had no identifiable markings.

One hundred and fifty participants completed the second phase study between January and April 2017 (Figure 3). Participants who participated in the first phase were not eligible to participate in the second phase.

#### 2.3.2. Measures

Body weight, height, BMI, and demographic information were measured and collected as described in the first phase. Participants were escorted to a private room with empty formula bottles, pitchers of water, a measuring cup, a Wi-Fi enabled tablet to access outside resources, and the infant formula container per the randomization schema. Two senior research staff gave oral explanations to all study participants using an IRB-approved script. These two research staff explained to participants that they would be escorted to a private room with all the materials needed to prepare 8 bottles of formula of varying serving sizes (two bottles each of 2, 4, 6, and 8 fl oz). These research staff handed the participant a sheet of paper with the randomized order in which they were to prepare the bottles. Participants were instructed to pass the bottles through a small opening to another room to be weighed, but not to speak with the research staff receiving the bottles. The research staff receiving the bottles, who were blinded to bottle preparation order, silently weighed each bottle after powder and water were added (OHAUS, Parsippany, NJ, USA). The research staff weighing the bottles were trained and certified to conduct the weighing procedures by the study coordinator prior to study initiation.

#### 2.3.3. Statistical Analysis

The randomization schema and statistical analyses were completed by the biostatistician using SAS/STAT^®^ software, (SAS System for Windows v9.4, Cary, NC, USA). All tests were completed with a significance level of α = 0.05 and β = 0.9, and findings were considered significant when *p* < α. Sample size estimates were based on a recent laboratory study, where participants were asked to prepare three randomly ordered sets of infant formula bottles (2, 4, 6, and 8 fl oz) prepared from formula powder [17]. The proportion of bottles with over-scooped powdered formula was 80%, and we hypothesized that the proportion of over-scooping would be reduced to 50% with modification to the preparation instructions. Seventy-five participants in each group were enrolled to detect a significant difference. The primary outcome was percent error (or accuracy) of powdered formula scooping in comparison to the expected gram weight described for the standard manufacturer instructions. Thus, the percent error, (measured-expected)/expected × 100, was used to analyze differences in formula powder, water, and caloric density (kcal/fl oz) separately with mixed-effect linear models accounting for repeated measures between participants by using an unstructured covariance matrix between measures. Results were adjusted for the following a priori-selected covariates: BMI, income, education level, age, and caregiver status. Least square means were used to report the effects of the treatment and bottle size, with *p*-values based on two-sample *t*-tests. For secondary analysis, a generalized linear mixed model was used to determine equivalency based on a priori bounds: 1%, 2%, 5%, and 10%. Equivalence was confirmed if the 90% confidence interval around the mean difference between the measured weights and expected weights was contained within the respective bounds. Covariates for the final model were selected for this model using risk estimation by investigating the association between each covariate falling within the percent error bounds using a Chi-squared test.

## 3. Results

### 3.1. First Phase—Participatory Research

Combined focus group participant demographic characteristics are summarized in Table 1. Focus group feedback on the infant formula instructions is summarized in Table 2 and Figure 4. Figure 4, Panel A shows the standard manufacturer instructions used and Panels B, C, D and E show modified labels per the feedback of the iterative focus groups. Focus group participants used the mixing guide first when asked to prepare infant formula and recommended moving the mixing guide to the top of the instructions as well as making the guide made larger. Participants appreciated the need for a combination of pictures and words to better understand the mixing guide and step-by-step preparation instructions. Participants found the safety information too long and not helpful for preparing the bottle. They recommended that the safety information be shortened, bulleted and moved to an inside panel similar to an over-the-counter medication label. Participants recognized the important role step 2 played in bottle preparation and recommended that the meaning of “level” and “unpacked” be clarified and the step expanded. To aid in bottle preparation, participants recommended a link to an instructional video created by a reputable source be included on the label.

### 3.2. Second Phase—Randomized Controlled Trial

Participants (Table 1) were 38.5 ± 16.9 years old, mostly female with some college education and defined as non-caregiver. Participant characteristics did not differ between individuals in the standard manufacturer and modified instructions groups. Participant characteristics did not differ between the two phases except for BMI as a categorical variable (*p* < 0.01).

#### 3.2.1. Powered Formula Dispensing

Participants who were assigned the modified instructions were able to dispense the powdered formula more accurately to the expected gram weight than participants who used the standard manufacturer instructions (−0.67 ± 0.76 vs. −4.66 ± 0.74% error; *p* < 0.0001; Figure 5).

Overall, accuracy in powdered formula dispensing was influenced by bottle size (*p* = 0.02) but not caregiver status (*p* = 0.18), BMI (*p* = 0.17), education level (*p* = 0.75), and income (*p* = 0.7), or age (*p* = 0.89). Percent error was different between the 2 and 4 fl oz (−2.11 ± 0.61 vs. −3.29 ± 0.6%; *p* = 0.03) and the 2 and 6 fl oz (−2.11 ± 0.61 vs. −3.79 ± 0.61%; *p* = 0.003) bottles, but no differences were observed between other bottle size comparisons. 

The proportion of bottles with powder dispensing within 1% accuracy differed between the instruction groups (modified: 16.2 vs. 6.8%; *p* < 0.0001). A similar degree of accuracy was observed for 2% bounds (33.3 vs. 18.2%; *p* < 0.0001), 5% bounds (65 vs. 46.2%; *p* < 0.0001), and 10% bounds (85.2 vs. 77.5%; *p* = 0.0003). A higher proportion of bottles made by individuals who identified as White were accurately dispensed compared to other study participants in all percent error bounds tested (1, 2, 5, and 10%; *p* < 0.0001 for all). Individuals who identified as a non-parental caregiver or individuals who had no experience preparing formula were more likely to dispense powder within 1% (*p* = 0.02 for both), 2% (*p* < 0.0001 and *p* = 0.02, respectively), and 5% (*p* = 0.02 and 0.01, respectively) bounds than their respective counterparts. A larger proportion of bottles made by individuals who did not work full time or earned less than $80,000/year contained powder within 1% error bounds (*p* = 0.004 and 0.01, respectively).

#### 3.2.2. Water Dispensing

There was no difference in percent error of water measurement between the instruction groups (standard: −1.4 ± 0.6 vs. 0.7 ± 0.6%; *p* = 0.38; Figure 5). Error in water dispensing did differ between bottle size (*p* < 0.0001) and education (*p* = 0.02), with differences observed between those whose highest level of education was some college/college degree and those with a high school diploma/general equivalency diploma (0.3 ± 1.0 vs. 3.0 ± 3%; *p* = 0.01).

The proportion of bottles with water dispensed within 1% accuracy did not differ between the preparation instruction groups (modified: 22.3 vs. 19%; *p* = 0.08) nor within 2% accuracy (38.5 vs. 35%; *p* = 0.1). Group differences were observed in the 5% bounds (65 vs. 58%; *p* = 0.006) and 10% bounds (79.2 vs. 73.3%; *p* = 0.009). Regardless of preparation instruction group, a higher proportion of bottles had water dispensed within 1% (*p* = 0.01), 2% (*p* < 0.0001), and 5% (*p* = 0.01) accuracy when they were made by individuals who reported a household income >$80,000. A higher proportion of bottles prepared by individuals who identified as a parent or individuals who reported previous experience in preparing formula contained water accurate within 2% (*p* = 0.01 and 0.03, respectively) and 5% (*p* = 0.0002 and 0.04, respectively) bounds than their respective counterparts. Females measured water more accurately than males within 5% and 10% error bounds (*p* = 0.03 and 0.002, respectively).

#### 3.2.3. Caloric Density

Caloric density was more accurate in the modified instruction group compared to the standard manufacturer instruction group (−0.3 ± 0.6 vs.−2.9 ± 0.9%; *p* = 0.03; Figure 5). Caregiver status was the only significant covariate identified, with non-caregivers being the most inaccurate and secondary caregivers being the most accurate (−4.2 ± 1.2 vs. 2.2 ± 2.0 vs. 1.8 ± 2.0%; *p* = 0.005).

## 4. Discussion

Iterative focus groups in the first phase provided guidance to modify standard manufacturer instructions for powdered infant formula to improve the dispensing accuracy of powdered infant formula in the second phase, a double-blind, randomized controlled trial. Majority of the recommended modifications made to the instructions focused on decreasing text and improving graphics especially related to powder dispensing, which translated to increased accuracy in powder dispensing in the modified instruction group. There was no difference in water dispensing between the two groups and, thus, caloric density remained more accurate in the modified instruction group.

A strength of this study was the inclusion of both caregivers and non-caregivers during both phases. This allowed for examination of differences in bottle preparation behaviors to make the study findings more generalizable, and to ensure the findings would not be biased by behaviors of caregivers who are experienced in infant bottle preparation. When data from both instruction groups were combined, participant characteristics including no or low experience in bottle preparation, identifying as a caregiver but not a parent, and having a lower income may indicate that these individuals followed the modified instructions more closely, leading to a greater proportion of bottles with accurately dispensed formula. Participant characteristics associated with a greater proportion of bottles with accurate water dispensing such as caregivers, female, past experience, and income may be related to experience with and competency in using a measuring cup or using the measurement markings found on formula bottles. While there were significant differences in percent error within bottle size, there may be limited clinical relevance to the difference in error between bottle sizes. Indeed, a difference of approximately 1% between smaller and larger formula portions would only account for a daily increase of 3–7 kilocalories, dependent on infant feeding patterns. In our previous study, when participants in the laboratory used standard manufacturer preparation instructions, 80% of the bottles prepared contained more than the recommended amount of powdered formula leading to a potential overfeeding of approximately 11% of daily energy needs [8]. In the present study, we did not observe this level of overfeeding. While a change in the direction of error was observed in this cohort, we were able to show a reduction in the magnitude of error with modified instructions. Individuals utilizing standard formula preparation instructions under dispensed by nearly 5%. Modeling this information from daily requirements of infants, this amount of formula error in an exclusively formula-fed infant would amount to a deprivation of 35% of caloric intake per week. Discrepancies in the direction of error between the two studies may be due in part to the present population being older. Additionally, participants were aware of the nature of the study, focusing on the accurate measurement and preparation of infant formula, which may be influenced participants to be more cognizant of overfeeding.

Similar to medications, infant formula, especially in the most common powdered form, requires label instructions and dosing instruments for appropriate measurement and usage. The U.S. Pharmacopeia has labeling recommendations for medications that take into account health literacy, but these have not been universally adopted [14] and do not extend to infant formula. Others have recommended label and instrument modifications and in-person medication counseling to decrease medication dispensing errors by being mindful of health literacy [15,16,18]. Currently, infant powdered formula instructions are governed by the Code of Federal Regulations Title 21 Chapter 1 Part 107 [19], and are on average at college level for reading difficulty [3]. The current study demonstrates the need for instruction revision to improve user understanding and the accuracy of powdered formula dispensing, particularly since formula use is most prevalent among families with low levels of education and poverty.

Breastfeeding initiation and support classes are rightfully found at local hospitals and physicians’ offices; however, there are no formal classes for those who need to formula feed or those who want to be prepared for formula feeding [3,4]. While the need for such classes was disputed in the focus groups of the first phase of the current study, all agreed that video instructions accessible on the web at any time of day would likely be helpful to new caregivers. Per focus group recommendation, an instructional video was created by study staff, and a link was included on the modified instructions. However, in the second phase, only two out of 75 individuals (2.7%) reported using the link to aid in their understanding of the written instructions.

The need for formula preparation instructions and therefore concern for over or under dispensing would be nearly eliminated with the adaptation of infant formula as a pre-portioned food. Commercial powdered formula does indeed exist in pre-portioned packets to yield a 4 fl oz serving but is positioned and priced as a product of convenience. Packaging and distributing infant formula as a pre-portioned food would systematically minimize the variability of the powdered formula weight that has been observed in this study and by others [6,7]. As evidenced in the discussions in the focus groups of the first phase, the weights of scoops of formula are directly affected by what instruments are used to measure the ingredients (i.e., infant bottles, scoops, measuring spoons, and measuring cups) and what the preparer considers most important on the label. In addition to the increased accuracy of using pre-portioned packs universally, the convenience benefit of pre-portioned formula packs would ensure that the infant received adequate formula even in the middle of the night or “on the go”. Furthermore, it would increase the ease for caregivers and clinicians to quantify infant food intake which is a notoriously challenging parameter to quantify [20].

This study clearly demonstrates utility of a participatory research model that led to changes in readability and understanding of infant formula instructions, which in turn increased the accuracy of formula dispensing and caloric density of bottles when compared to standard manufacturer instructions in a laboratory randomized controlled trial. We acknowledge the limitation of testing the accuracy of one manufacturer instruction; however, reading level across the major infant formula brands is similar [3]. In addition, the education level in the random population tested was not particularly low and the results may differ in a population with lower health literacy. The bottles were not prepared for infant consumption and bottle preparation may differ in a free-living setting. Indeed, the role of alternative strategies including improved labeling, the use of a universal label across all brands, increased access to instructional videos and cost-effective pre-portioned formula packets on long-term provision of infant nutrition and growth remains to be tested.

## Figures and Tables

**Figure 1 nutrients-12-01150-f001:**
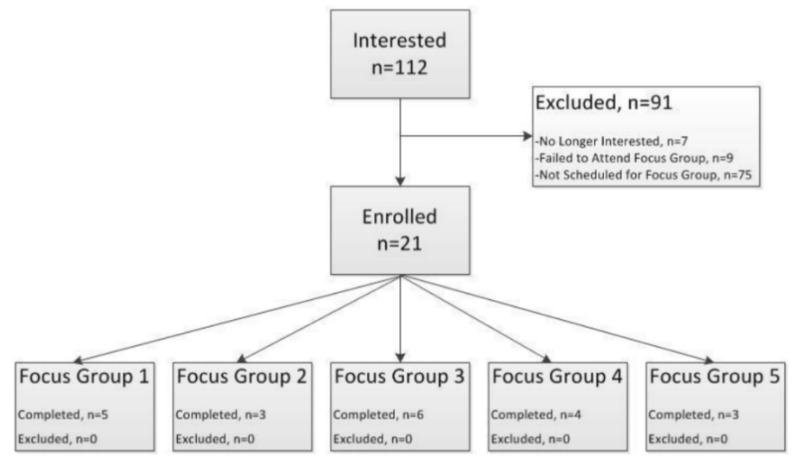
Participatory Research Phase Consort Diagram.

**Figure 2 nutrients-12-01150-f002:**
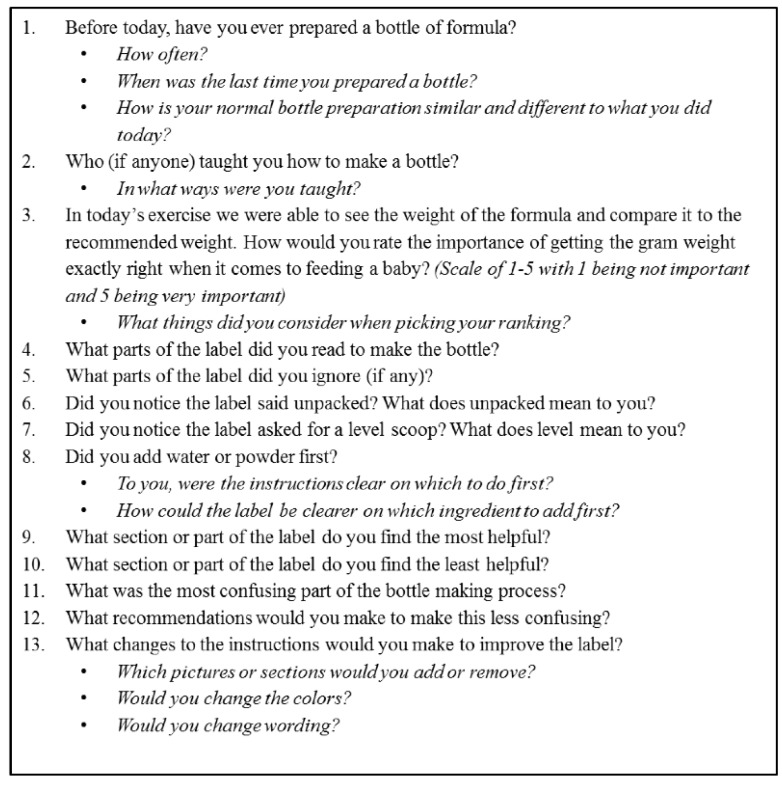
Focus Group Questions.

**Figure 3 nutrients-12-01150-f003:**
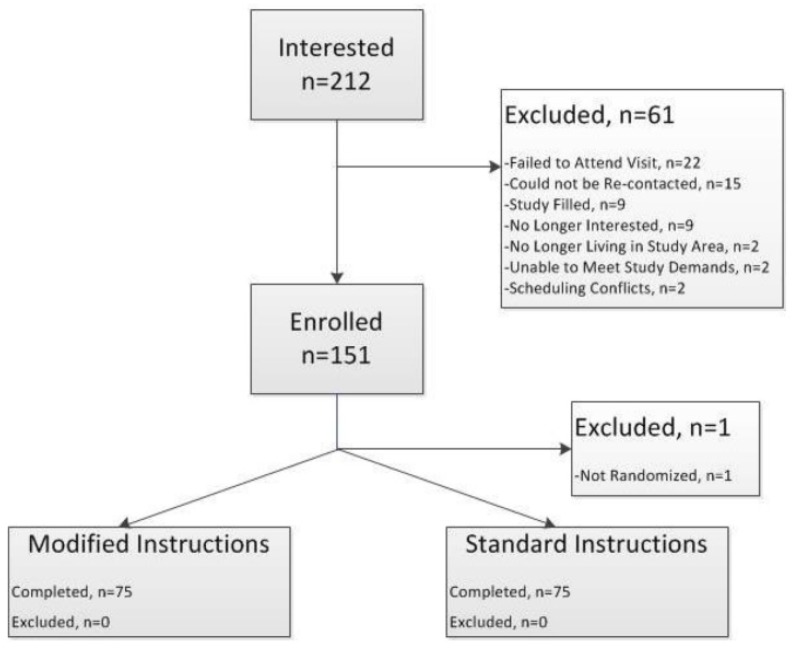
Randomized Control Trial Consort Diagram.

**Figure 4 nutrients-12-01150-f004:**
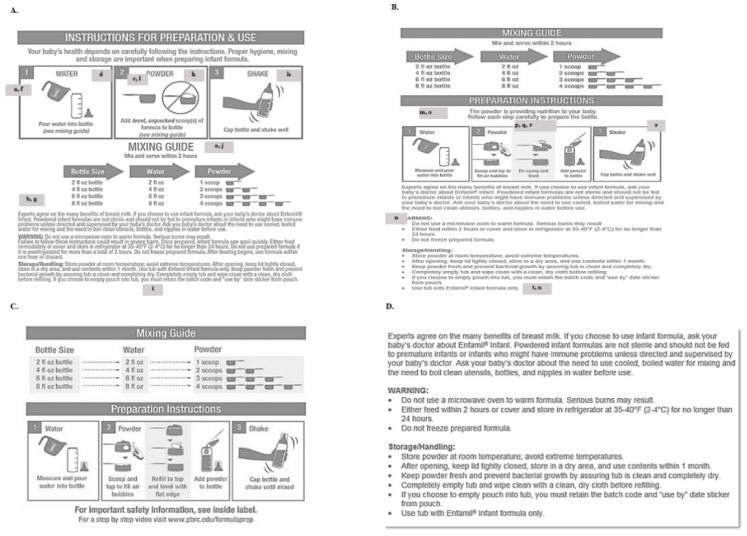
Modifications to Infant Formula Preparation Instructions. Panels (**A**–**D**) show modified labels per the feedback of the iterative focus groups. Panels (**C**,**D**) would be on the same label with panel (**D**) on the inside flap. Lower case letters indicate the primary recommendations by each focus group as summarized in Table 2.

**Figure 5 nutrients-12-01150-f005:**
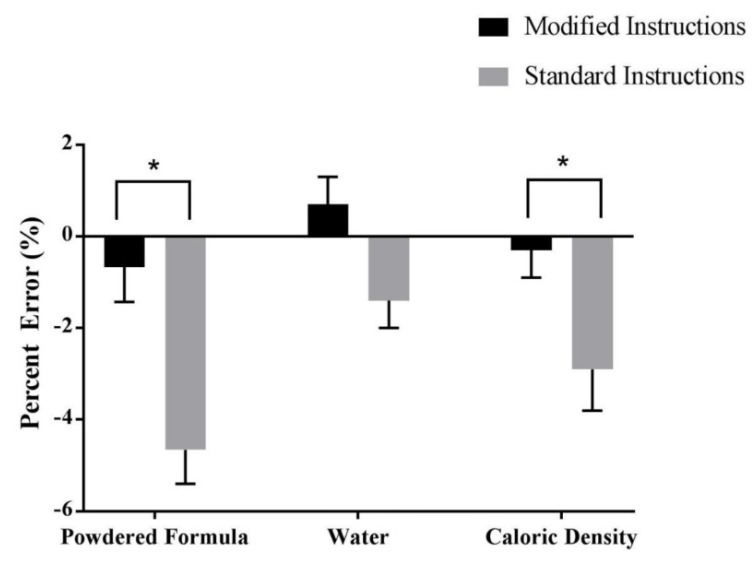
Percent Error of Powdered Formula Dispensing, Water Dispensing, and Caloric Density for the modified instructions group and the standard instructions group. Data are presented as the mean ± SD and * denotes significant (*p* < 0.05) difference between groups. Black bars denote the modified instructions group and gray bars denote the standard instructions group.

**Table 1 nutrients-12-01150-t001:** Characteristics of Study Participants.

	All Phase 1 (*n* = 21)	Phase 2 Modified Instructions (*n* = 75)	Phase 2 Standard Instructions (*n* = 75)
Age, year	32.5 ± 11.7	40.6 ± 17.6	36.0 ± 15.8
BMI, kg/m^2^	26.8 ± 8.1	29.5 ± 7.9	29.6 ± 6.6
BMI Classification, *n* (%)			
Normal or Under Weight	14 (66.7)	24 (32.0)	18 (24.0)
Overweight	2 (9.5)	20 (26.7)	23 (30.7)
Obese	5 (23.8)	31 (41.3)	34 (45.3)
Race, *n* (%)			
Caucasian	15 (71.4)	54 (72.0)	46 (61.3)
African American	6 (28.6)	16 (21.3)	23 (30.7)
Other	0 (0.0)	5 (6.7)	6 (8.0)
Gender, *n* (%)			
Male	4 (19.0)	16 (21.3)	10 (13.3)
Female	17 (81.0)	59 (78.7)	65 (86.7)
Caregiver of Infants Status, *n* (%)			
Primary Caregiver	6 (28.6)	10 (13.3)	15 (20.0)
Secondary Caregiver	5 (23.8)	11 (14.7)	10 (13.3)
Non-Caregiver	10 (47.6)	54 (72.0)	49 (65.4)
No Answer	0 (0.0)	0 (0.0)	1 (1.3)
Education, *n* (%)			
High School Diploma/GED or Less	1 (4.8)	5 (6.7)	8 (10.7)
1–3 Years College	5 (23.8)	37 (49.3)	26 (34.7)
College Degree	9 (42.8)	17 (22.7)	25 (33.3)
Post-Graduate Degree	6 (28.6)	16 (21.3)	16 (21.3)
Income, *n* (%)			
<$30,000/year	5 (23.8)	21 (28.0)	21 (28.0)
$30,000–$99,999/year	9 (42.9)	28 (37.3)	32 (42.7)
>$80,000/year	7 (33.3)	25 (33.3)	21 (28.0)
No Answer	0 (0.0)	1 (1.3)	1 (1.3)
Employment, *n* (%)			
Full Time	11 (52.4)	22 (29.3)	33 (44.0)
Not Full Time	10 (47.6)	53 (70.7)	42 (56.0)
Parent/Guardian, *n* (%)			
Yes	11 (52.4)	38 (50.7)	43 (57.3)
No	10 (47.6)	37 (49.3)	32 (42.7)
Prepared Formula in Past Year, *n* (%)			
Yes	8 (38.1)	21 (28.0)	26 (34.7)
No	13 (61.9)	54 (72.0)	49 (65.3)

**Table 2 nutrients-12-01150-t002:** Focus Group Recommendations.

Focus Group Order	Participant Characteristics	Primary Recommendations *
First	*n* = 5 24.6 ± 3.8 years old [range, 22–32] Female Caucasian College educated Mixed caregiving experience	Pictures were helpful Safety and handling information was too lengthy and not helpful (bullet important information) Meaning and importance of “level” and “unpacked” is not clear Enlarge main text
Second	*n* = 3 31.3 ± 11.0 years old [range, 18–45] 2 males and 1 female African American ≥High school diploma Mixed caregiving experience	e. Mixing guide was helpful (move to top) f. Pictures were helpful g. Safety and handling information was too lengthy and not helpful (bullet important information) h. Only picture a correctly leveled scoop i. Provide instructional video accessible by QR code
Third	*n* = 6 33.7 ± 3.7 years old [range, 29–38] Female 5 Caucasian and 1 African American ≥College degree All caregivers	j. Mixing guide was helpful (move to top) and increase size k. Step 3 “shake” and picture was not helpful l. The use of “level” and “unpacked” is not clear (equipment or dispensing modifications)
Fourth	*n* = 4 29.0 ± 5.5 years old [range, 21–35] Female Caucasian Mixed education Non-caregivers	m. Pictures were helpful n. Move safety information to inside flap o. Add bottle cap to pictures p. The use of “rescoop” was confusing q. Instruction to level was clear (add with flat edge; edit flat edge picture) r. Divide step 2 into two steps s. Keep step 3, “shake” t. Video should be made by credible group—not by formula company u. Put web address instead of QR code
Fifth	*n* = 3 49.3 ± 18.4 years old [range, 24–67] 1 male and 2 females 1 Caucasian and 2 African American Mixed education Non-caregivers	v. Unanimously approved the instructions for second phase testing w. Mixing guide should be first x. Combination of words and pictures y. Liked “level with flat edge” z. Agreed safety information on inside flap, but consider moving important safety information to front

* Primary recommendations are also denoted (lower case letters) on the infant formula instructions presented in Figure 1B–D.

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
