# Peer review of "Modifications to Infant Formula Instructions Improve the Accuracy of Formula Dispensing"

_nutrients, 2020, doi:10.3390/nu12041150_

Round 1

Reviewer 1 Report

This well-written paper presents an interesting topic, and it potentially has a moderate public health impact. There are a few strengths with this study, for example the two phases of the project in which the participatory research informed the randomized controlled trial. However, some limitations of this study considerably dampened the significance and its contribution to the field.

Systematically, participants of both groups in the current RCT under-dispended the powdered milk. In contrast, a paper published by the same laboratory in 2019 (Altazan et al.) demonstrated that only 3% of the total amount of bottles prepared by participants were under-dispensed, and a whopping 78% of the bottles were over-dispensed. Due to these original findings, there is a great rationale to conduct the current study- to develop infant formula preparatory instructions that are easier to read and to follow in order to prevent over-dispensing that can lead to overfeeding. Using a similar error cutoff of ± 5% of the expected gram weight (or 5% bound) in Altazan et al., 2019, we can assume that about 50% of the participants in the “standard instructions” group under-dispensed the powdered milk (53.8% total did not meet the 5% accuracy cut point, presumably this included over and under-dispensing) in the current study. Due to this discrepancy, there are some major concerns with this study:

  1. Participants recruited for the current RCT might not be representative of individuals who would benefit from the instructions modification- individuals who are or will be caring for infants. Compared to the participants in Altazan et al., 2019, the current cohort of participants were older (38.5 y vs. 29 y) and only 30.6% (vs. 52.8%) of them were considered a caregiver (primary or secondary). There are benefits of recruiting non-caregivers for the study, as they might potentially be caregivers in the future; however, with a majority of older females in this cohort, this possibility might be low (they might act as secondary caregiver).
  2. Table 1 should provide the demographics of participants in the modified vs. standard group separately. It is important to examine if there are any significant differences in characteristics between groups.
  3. Authors did not provide sufficient information on study procedures for the RCT phase.
  4. a) How were participants being recruited? This might explain why a cohort of non-representative individuals were recruited- a convenience sample.
  5. b) Did the recruitment materials include information about assessing the accurate measurement and preparation of infant formula, which has led to biases in powdered milk dispensing?
  6. c) What is the setting? Did participants prepare the formula in a lab or kitchen area? What is the laboratory set up?
  7. c) What instructions were given to participants? Did the same research staff member conduct all of the appointments? Were scripted phrases used across all participants to minimize variability?
  8. Many limitations of the study were not discussed by the authors in the Discussion section.

In summary, despite the limitations listed above, this project did successfully demonstrate that improved instructions can help to increase the accuracy of powdered milk dispensing among non-caregivers (predominately). Therefore, this project can be strengthened by recruiting a small cohort of subjects who are more representative of the typical caregivers or future caregivers to test for the modified instructions derived from their participatory phase. With a small cohort, the most ideal study design is a randomized crossover design with a few days of washout period between each test group.

Reviewer 2 Report

This is a novel study to evaluate a strategy for improving the accuracy of formula dispensing. The combination of focus groups for input on the design and RCT to compare dispensing under standard versus modified formula dispensing instructions is particularly unique. I have only a few questions:

Abstract: include measurement of accuracy, because the results for powdered formula dispensing are not interpretable (beyond significance) without the definition.

Lines 71-78: Focus group participants. It would be helpful to understand how participants were recruited and to highlight that the participants were drawn from the overall population. It was not until later in the manuscript did I understand who was included in the studies.

Lines 121-127: Design. Could the authors provide more details about where the study was conducted and the instructions given to the participants. For example, did the participants know the study was about accuracy of formula dispensing? The discussion section suggests that they knew that the study was about the accuracy of measurement, which could have biased the results. Were the instructions similar to those given in the original study that the authors reference? Without having more details about the instructions and information given to the participants, it is difficult to understand the differences in the results between studies.

Line 181: Table 1. Are the p-values actually relevant? Given the very small sample in the focus groups and the purpose of each phase of the study, I am not sure that the p-values add value.

Line 184: Table 2. The means and standard deviations for the participants ages are based on extremely small sample sizes (e.g., n = 3 to 6), so median and range would be more relevant. I have a similar comment for the mean BMI and age of the focus group in Table 1.

Line 192+:Results. I would like to see the results comparing the standard and modified groups in a table. While the figure provides high-level information, a table would be more informative. In the text, it is difficult to follow some of the numbers – so the difference in the percent error for the 2 oz and 4 oz bottles between the standard and modified group was -3.29% versus -2.11%? Additionally, are these differences clinically meaningful?

Line 257: Discussion. Does the comparison with bottles prepared in the laboratory having more than the recommended amount of powdered formula refer to bottles prepared in another study or bottles prepared by the manufacturer?

Round 2

Reviewer 1 Report

The authors have been highly responsive to the earlier reviews. This revised manuscript is much improved. It is likely to be of considerable interest to pediatricians and those interested in pediatric research as well as the general readership of Nutrients. I only have one comment on the method:

The below response is a more thorough explanation of the study method.

Two senior staff gave oral explanations to ALL study participants using an IRB approved script. These two researchers explained to participants that they would be escorted to a private room with all the materials needed to prepare 8 bottles of formula of varying sizes. These research staff handed the participant a sheet of paper with the randomized order in which they were to prepare the bottles. Participants were instructed to pass the bottles through a small opening to another room to be weighed, but not to speak with the research staff receiving the bottles. The research staff receiving the bottles, who were blinded to bottle preparation order, silently weighed each bottle after powder and water were added. The research staff weighing the bottles were trained and certified to conduct the weighing procedures by the study coordinator prior to study initiation.

Author Response

We thank Reviewer 1 for all the previous comments and suggestions that have significantly enhanced this manuscript. We agree that the suggested paragraph is a more thorough explanation of the study method and have replaced the text. We applied minor edits to the paragraph including: the addition of bottle serving sizes, the source of the scale for bottle weights, and edited titles of the research staff for consistency.

Reviewer 2 Report

The authors clearly addressed my concerns in the revised manuscript. This manuscript will make an important contribution to the literature, and I have no additional comments regarding this manuscript.

Author Response

We thank Reviewer 2 for all the previous comments and suggestions that have significantly enhanced this manuscript.